# Using Training Samples Retrieved from a Topographic Map and Unsupervised Segmentation for the Classification of Airborne Laser Scanning Data

**Zhishuang Yang [1], Wanshou Jiang [1],* , Yaping Lin [2] and Sander Oude Elberink [2]**

[1]  State Key Laboratory of Information Engineering in Surveying, Mapping and Remote Sensing, Wuhan University, Wuhan 430079, China; tcyzs@whu.edu.cn

[2]  Faculty of Geo-Information Science and Earth Observation (ITC), University of Twente, 7514 AE Enschede, The Netherlands; y.lin@utwente.nl (Y.L.); s.j.oudeelberink@utwente.nl (S.O.E.)

*  Correspondence: jws@whu.edu.cn; Tel.: +86-27-68778092 (ext. 8321)

**Abstract:** The labeling of point clouds is the fundamental task in airborne laser scanning (ALS) point clouds processing. Many supervised methods have been proposed for the point clouds classification work. Training samples play an important role in the supervised classification. Most of the training samples are generated by manual labeling, which is time-consuming. To reduce the cost of manual annotating for ALS data, we propose a framework that automatically generates training samples using a two-dimensional (2D) topographic map and an unsupervised segmentation step. In this approach, input point clouds, at first, are separated into the ground part and the non-ground part by a DEM filter. Then, a point-in-polygon operation using polygon maps derived from a 2D topographic map is used to generate initial training samples. The unsupervised segmentation method is applied to reduce the noise and improve the accuracy of the point-in-polygon training samples. Finally, the super point graph is used for the training and testing procedure. A comparison with the point-based deep neural network Pointnet++ (average *F1* score 59.4%) shows that the segmentation based strategy improves the performance of our initial training samples (average *F1* score 65.6%). After adding the intensity value in unsupervised segmentation, our automatically generated training samples have competitive results with an average *F1* score of 74.8% for ALS data classification while using the ground truth training samples the average *F1* score is 75.1%. The result shows that our framework is feasible to automatically generate and improve the training samples with low time and labour costs.

**Keywords:** automatic training samples generation; unsupervised segmentation; graph convolutional neural network; ALS point clouds

## 1. Introduction

In recent years, the use of point clouds has attracted wide attention in computer vision, photogrammetry, and remote sensing. Point clouds as a source data can be used to process digital terrain models (DTM), landscape models, and three-dimensional (3D) city models. Supervised algorithms based on their capability of understanding the relationship between data and labels are proposed for the classification work. Adequate training samples are needed to train supervised networks. Most of the point clouds training samples are manually labeled which is time-consuming. Unsupervised methods seem to be an excellent solution to generate training samples. Some unsupervised methods [1,2] have been proposed to label indoor point clouds and have achieved good performances. However, for outdoor space, especially large scale urban scenes, it is difficult for unsupervised methods to classify objects correctly. Most of the unsupervised methods for outdoor space point clouds are used to classify

a single category [3–5] such as terrain or trees. Methods that can classify multiclass [6–9] are only suitable for simple scenarios, or they miss complex classes such as bridges [10]. These methods are especially sensitive to noise and different backgrounds [11]. In different urban scenes, unsupervised methods need different parameters. For the purpose of generating training samples in large scale outdoor space, a more general method should be proposed.

The two-dimensional (2D) topographic map contains locations of objects and various corresponding class labels, and the 2D topographic map also has little fluctuation on different terrain scene. To generate sufficient training samples for deep learning methods on the large urban scene and reduce labour costs in manual annotation, an automatically generated training sample framework is proposed in this paper. The scientific contributions are as follows:

- We propose a point-in-polygon operation using a 2D topographic map to generate the initial training samples automatically;
- We modify the initial training samples using an unsupervised segmentation method; the unsupervised strategy reduces the noisy effect and improves the accuracy of our point-in-polygon training samples;
- We use the intensity value to improve the segmentation performance on small categories and use a graph convolutional neural network for the training and testing work.

The rest of this paper is organized into four additional sections. The related work to this subject is discussed in Section 2. Section 3 introduces our methodology. We present the experimental results in Section 4. These results are discussed in detail in Section 5. We provide concluding remarks and suggestions for future work in Section 6.

## 2. Related Work

For point clouds classification, the supervised method is divided into the following two categories: the point-based method and the segment-based method.

With respect to the point-based methods, Lodah et al. [12] applied Adaboost to classify ALS point clouds into four categories. A new metric, pseudo error, was used to measure the performance of hypotheses and five features were used in the classification work. Chehata et al. [13] applied random forests (RF) to label the point clouds due to its accuracy and efficiency. Backward elimination of features was used to analyze the relevance of each lidar features. Mallet et al. [14] applied two waveform processing methods to extract the full-waveform features of the lidar point clouds. A SVM classifier was used to label the point clouds into building, ground, and vegetation points. To increase the distinctiveness of geometric features and improve the classification accuracy, Weinmann et al. [15] presented a framework composed of neighborhood selection, feature extraction, feature selection, and classification. Seven neighborhood definitions, 21 geometric features, seven approaches for feature selection, and 10 classifiers were used in this framework. Niemeyer et al. [16] integrated an RF classifier into a pairwise conditional random fields (CRF) framework. The consideration of contextual information and the usage of a large number of features improved the labeling result. For the method, its pairwise potentials were based on a Potts model. This led the interactions to only occur at a very local level and caused misclassification on some isolated clusters of points. Many researchers have proven that the CRF method can handle these long-range interactions. Luo et al. [17] presented a multi-rand and asymmetric conditional random fields (maCRF) method in which prior information of scene-layout compatibility was applied for the long-range dependency problem. The maCRF consisted of two models, one used for enhancing the local labeling smoothness within a short-range and the other used for favoring the arrangement between different objects within a long range. The final classification results were obtained from the combination of these two models. Another solution was proposed by Xiong et al. [18], a multistage inference procedure was used to capture the contextual relationships between 3D points. In this method, point clouds statistics were trained and the relational information was learned over pointwise and segment-wise scales.

Due to their capability of feature learning, deep convolutional neural networks (CNN) have achieved superior performance in many fields. In point-based classification work, convolutional neural networks also consider contextual information. Most current approaches operate in an ordered regular space. Tosteberg et al. [19] proposed a method that projected the point clouds into 2D RGB images via Katz projection and a pretrained CNN was used to classify the images. Boulch et al. [20] presented a method using a convolutional neural network on snapshots of the point clouds. First, red-green-blue views and depth composite views were generated by suitable snapshots of point clouds. Then, pixel-wise labeling was performed using the fully convolutional network (FCN). Point clouds were labeled by a fast back-projection process in the 3D space. Yang et al. [21] presented a point-based feature image generation for CNN. Each point was transformed into a feature image according to the geometric features in its neighbor. The CNN model learned relationships between the point and the feature image. Thomas et al. [22] presented a new design of point convolution that operated on point clouds without any intermediate representation. The deformable strategy was used to adapt the local geometry of kernel points. The regular subsampling method was used to improve the efficiency and the robustness to varying point densities. Another solution has been to operate on the unstructured point clouds directly. Qi et al. [23] presented a novel type of neural network that effectively respected the permutation invariance of points in the input called Pointnet. Furthermore, in [24] he introduced a hierarchical neural network that applied PointNet recursively on a nested partitioning of the input point set. By exploiting metric space distance, this framework, called Pointnet++, could learn deep point set features efficiently and robustly, thus, improving the semantic labeling results. The Pointnet and Pointnet++ had a limit receptive filed over the 3D scene, and Engelmann et al. [25] presented two extension strategies to enlarge it. The neighborhoods of point clouds contained various features that were useful for classification.

Segment-based methods have also been used in supervised classification of the point cloud. After a segmentation, more stable features can be obtained since segments contain some extending features as compared with a single point within its local neighborhood. Serna et al. [26] presented a framework for classifying 3D urban objects. The watershed approach was used to separate the connected segments after ground segmentation. The support vector machine (SVM) was used for further labeling work. Hu et al. [27] combined unsupervised segmentation and a supervised region growing method to learn reliable contextual information. To avoid the over- and under-segmentation, an effective multiscale processing method was used to perform joint patch segmentation and labeling. In [28], Niemeyer proposed a two-layer CRF. The first layer CRF was used to generate segments. The segments contained a larger-scale context and were introduced as an energy term for the next iteration of the next CRF layer. Vosselman et al. [29] combined different point clouds segmentation methods to obtain the best segmentation result which minimized both under-segmentation and over-segmentation. The, CRF was used to obtain the contextual segment-based semantic labeling result. The structure was capable of handling complex urban areas with a large number of categories. The combination of small and large segments produced by the hierarchical structure made the interaction between nearby and distant points come true. The contextual information was learned by using a CRF. The mix of small and large segments which allowed the interaction between nearby and distant points, thus, improved the final results. The segment-based strategies have also been used in deep neural networks. Yang et al. [30] proposed a three-step region growing segmentation for the ALS point clouds classification. Feature images were generated based on the segmentation result. The multiscale convolutional neural network was trained to automatically learn contextual information of each segment from generated feature images across multiple scales. Engelmann et al. [31] introduced two kinds of neighborhoods within the point clouds into the neural network. These neighborhoods helped to learn the regional descriptors. Two dedicated loss functions were used to further structure the point feature space. Landrieu et al. [32] presented a novel deep learning-based framework that tackled the challenge of semantic segmentation of large-scale point clouds of millions of points. Using a non-parametric segmentation model [33], he turned the 3D point clouds into an efficient structure called super point graph (SPG) to enrich the

representation of contextual relationships between object parts which were, then, labeled by the graph convolutional network.

Supervised algorithms have the potential to explore properties and combinations of properties that could not be found by humans. The downside is that a fair amount of training data is needed to let the algorithm learn the optimal parameters. Until now, it remains rather vague in practice how many training samples are needed to train a proper network. Most of the training data is manually labeled and it is time-consuming. Some unsupervised methods can be used to generate training samples. For indoor space, Endre et al. [1] applied the latent Dirichlet allocation(LDA) to identify the indoor laser scanning point clouds in an unsupervised way. The method, first, obtained well-segmented point clouds, then, the LDA was used to cluster the segment objects. For a more complex scene, Poux et al. [2] derived two feature sets from the coordinate and the voxel entities. A knowledge-based decision tree was proposed using these features for unsupervised labeling of the point clouds. A comparison with the state-of-the-art deep learning methods has shown that this unsupervised method achieved high *F1* score (>85%) for planar-dominant classes. For the point clouds of large-scale urban environments, accurately classifying objects using the unsupervised approaches is a difficult task. Some methods have been proposed for only classifying a single category, such as terrain or vegetation. Vosselman et al. [3] presented an erosion operator-like algorithm to generate DTM in an unsupervised way. The experiment showed that the filter effect degrades as the point density decreases. For a more complicated scene, an efficient mathematical morphology-based multilevel filter [4] was proposed by Chen et al. to generate DTM. Preliminary non-ground points were first identified with the characteristics of the ALS data. Then, the localized mathematical morphology opening operations were used on remaining points. Rutzinger et al. [5] applied an object-based point cloud analysis approach. This method combined segmentation and classification of ALS points and could detect the vegetation points in urban environments. Using 3D properties, a good separability of building and terrain points was achieved when they were occluded by vegetation. To distinguish more types of categories, Lafarge et al. [6] applied the Markov random field (MRF) based method to classify the point clouds into building, vegetation, and ground. Geometric 3D primitives combined with mesh patches represented the irregular roof components and the various urban components interacted through nonconvex energy minimization. Gerke et al. [7] applied both ALS point clouds and corresponding images for unsupervised classification work and four objects, i.e., buildings, trees, vegetated ground, and sealed ground were detected. The method was embedded in MRF, and graph-cut was used for energy optimization. Sun et al. [8] applied the graph-cut-based method to detect vegetation. Hierarchical Euclidean clustering was used to extract ground and building rooftop patch. After a "divide-and-conquer" refinement, mesh models could be generated. Rau et al. [9] classified the point clouds generated from oblique aerial imagery and vertical aerial imagery into five classes. A rule-based hierarchical semantic classification scheme was proposed.

These unsupervised methods that classify multiclass are sensitive to noisy and different backgrounds, or they can miss some complex classes. In this paper, a framework using 2D topographic maps and an unsupervised segmentation step is proposed to handle these problems.

## 3. Methodology

### 3.1. Point-in-Polygon Operation

The workflow of point-in-polygon operation is shown in Figure 1. The topographic map and its corresponding ALS data were used to generate initial training samples. Five categories (terrain, water, vegetation, building, and bridge) were selected in our training samples. These classes were also represented in the ground truth label from the same ALS dataset.

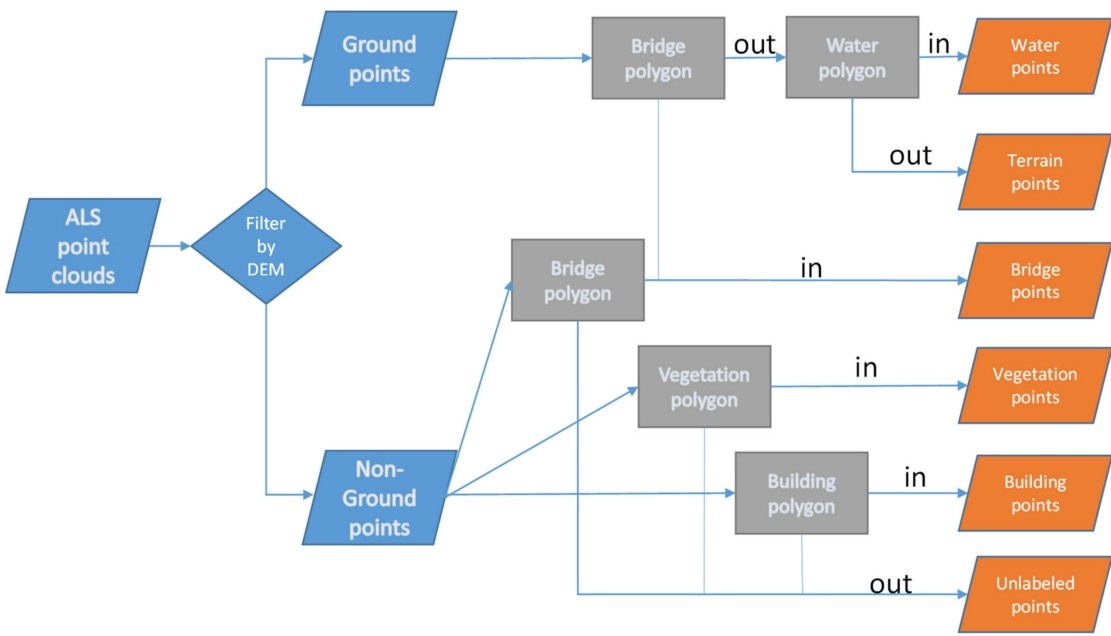

**Figure 1.** Workflow for the point-in-polygon operation.

The first step was to divide the input ALS point clouds into ground points and non-ground points by a DEM filter. DEM was generated by LAStools [34]. Then, the topographic map was used for further operation. Four polygon maps were derived from the topographic map. Each polygon map layer was assigned a class label. The point-in-polygon operation was performed on the entire ALS points. For the ground part, bridge polygon was applied at first. If a point was in the bridge polygon, the bridge label was assigned to it. If not, water polygon was used for further labeling. If the point was in the water polygon, the water label was assigned to it. If not, the terrain label was assigned to this point. For the non-ground part, the operation was simpler. Bridge, building, and vegetation polygons were used. If a point was in these polygons, the corresponding label was assigned to it. If not, the value "0" was assigned to this point, which means unlabeled. After the point-in-polygon operation, each point in the ALS data had a label.

Figure 2 shows the comparison between the ground truth result and the point-in-polygon result. As we only used four polygon maps from the topographic map, some of the points in the ALS data remain unlabeled (black colour points in Figure 2b). Most of the points are correctly labeled. However, in reality, there were several reasons why the initial labels did not work properly in Figure 3. First, one explanation is that there are situations where the labeling is not correct. Next, another explanation is that with a segmentation step we can solve many of these errors in the training data and, even then, if the training data is not perfect, we show later that the classification by super point graph is capable of dealing with those errors. As shown in Figure 3a, some building facade points and tree points remained unlabeled (in black). The registration between both datasets is not perfect. Since the 2D topographic map accuracy is in the order of 0.1 to 0.2 m, some points are unlabeled or mislabeled. Directly using the 2D map to assign labels to 3D point clouds definitively caused assignment errors, especially near borders of polygons. Some trees grow over the building roof. Points in these trees were assigned wrong labels since they are in the building polygons, as shown in Figure 3b. Objects could have been changed between the acquisition of the topographic map and the point clouds. The ALS data in our experiments was acquired later than the topographic map. Some houses had not been built when acquiring the topographic map, and therefore they remained unlabeled in the point-in-polygon training samples, as shown in Figure 3c. Not all the topographic features were used in our point-in-polygon operation. Vegetation objects in our 2D topographic maps were represented by point features and

polygon features. Since only polygon features were used in our framework, some tree points remained unlabeled, as shown in Figure 3d.

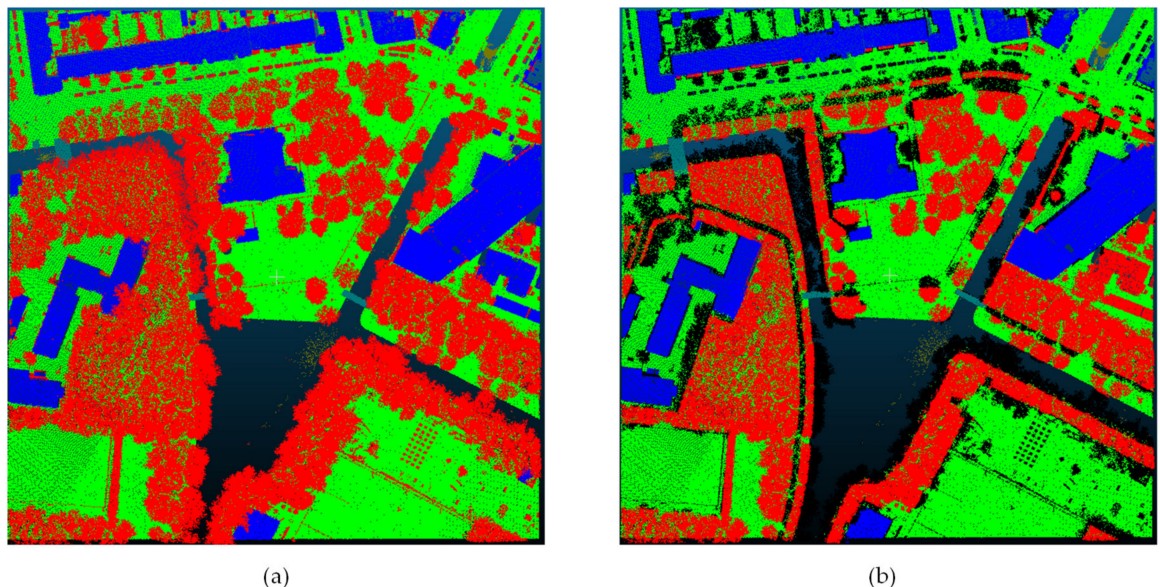

(a) (b)

**Figure 2.** (**a**) Ground truth result; (**b**) Corresponding point-in-polygon result. Building, navy blue; tree, red; water, brown; terrain, green; bridge, cyan; and unlabeled, black.

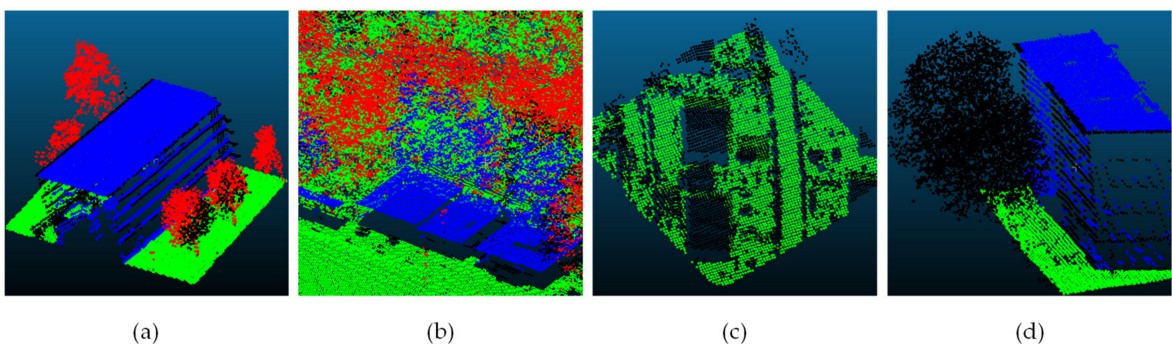

(a) (b) (c) (d)

**Figure 3.** Incorrect labels in the point-in-polygon results. (**a**) Unlabeled building and tree points caused by datasets registration problem; (**b**) Mislabeled tree points as they are in the building polygon; (**c**) Unlabeled building points caused by missing building polygons; (**d**) Unlabeled tree points caused by missing vegetation polygons. Building, navy blue; tree, red; terrain, green; and unlabeled, black.

### 3.2. Unsupervised Segmentation

After we obtained the training samples, an unsupervised segmentation method was used to make the input data into a simple shape and modify the initial training samples. What we expect for the segmentation result is that points in the cluster belong to the same categories. This part plays an important role in our framework. The unsupervised segmentation can refine some mislabeled or unlabeled points in our point-in-polygon operation. Considering the time efficiency and the segmentation accuracy, the global energy model proposed by Guinard et al. [33] was used. This global energy model helps the segmentation result adapt to the local geometric complexity. Simple shape categories such as terrain or water are in large clusters. Categories such as building and tree are in smaller components. The segmentation results are defined by the optimization of the global energy model:

$$\underset{g \in R^{d_g}}{argmin} \sum_{i \in C} \|g_i - f_i\|^2 + \mu \sum_{(i,j) \in E_{nn}} \omega_{i,j} \left[g_i - g_j \neq 0\right]. \tag{1}$$

where $i \in C$ denotes the point number in ALS data, $R^{d_g}$ are geometric features describing the point local neighbor shape, $G_{nn} = (C, E_{nn})$ is the 20-nearest neighbor adjacency graph, and $[\,]$ denotes the Iverson bracket. The edge weight $\omega$ is calculated according to the linearly decreasing of edge length and $\mu$ is the parameter that we could adjust to restrict the segmentation results coarseness.

Landrieu applied the $\iota_0$ cut pursuit [35] to solve the optimization problem since it can quickly find an approximate solution within a few graph cut iterations. Four features, i.e., linearity, planarity, scattering, and verticality are used to describe the local neighbor shape in his paper. The segmentation results are shown in Figure 4. The label is assigned according to the most frequent point label in each segment. Some unlabeled and mislabeled points (in Figure 3) are fixed, as shown in Figure 4a,b.

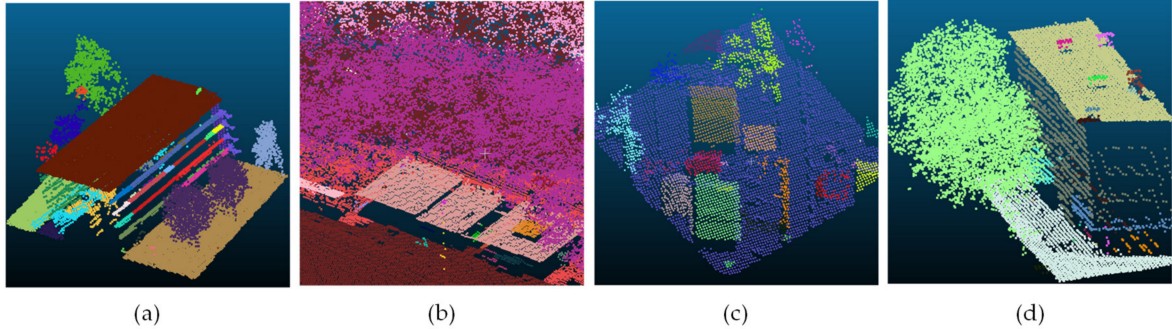

(a)　　　　　　　　　(b)　　　　　　　　　(c)　　　　　　　　　(d)

**Figure 4.** Unsupervised segmentation results. (**a**) Segmentation result on unlabeled area caused by datasets registration problem; (**b**) Segmentation result on mislabeled tree points as they are in the building polygon; (**c**) Segmentation result on unlabeled building points caused by missing building polygons; (**d**) Segmentation result on unlabeled tree points caused by missing vegetation polygons.

When it comes to the situation that two categories are adjacent in space and have similar geometric features such as terrain, water, and bridge, it is difficult to divide them by only using four geometric features, as shown in Figure 5a. The echo intensity values have different performance on different categories, which can solve the situation. As shown in Figure 5b, after adding the intensity value into the feature group, classes such as water, terrain and bridge can be separated accurately. Although some segments are unlabeled or mislabeled, as shown in Figure 3c,d, after the unsupervised segmentation, most of the segments have correct labels and can be used to train the super point graph.

### 3.3. Super Point Graph

To construct the super point graph $G_{sp} = (\mathcal{N}, \varepsilon)$, segments are treated as nodes, $\mathcal{N}$ (super point) and relations between neighbor segments are treated as edges $\varepsilon$. For the input point clouds, the symmetric Voronoi adjacency graph [36] $E_{vor}$ is calculated at first. The edges of the super point graph are defined if two super points S and T have at least one edge in $E_{vor}$:

$$\varepsilon = \left\{ (S, T) \in \mathcal{N}^2 \middle| \exists (i, j) \in E_{vor} \cap (S \times T) \right\} \tag{2}$$

The mean and deviation of edge offset values are calculated as edge features for the super point graph. Shape and size of neighbor super points such as centroid offset, length ratio, surface ration, volume ratio, and point count ratio values are also treated as super edge features. The shape features of super point S are calculated as follows:

$$\begin{aligned}
length(S) &= \lambda_1 \\
surface(S) &= \lambda_1 \lambda_2 \\
volume(S) &= \lambda_1 \lambda_2 \lambda_3
\end{aligned} \tag{3}$$

$\lambda_1 > \lambda_2 > \lambda_3$ are eigenvalues calculated form the points in each super point.

The label of a super point is represented by the most frequent label in its corresponding segment. After embedding every super point $S_i$ into a vector $z_i$ by using the Pointnet [23], an edge convolutional neural network is used for contextual segmentation. Landrieu applies the gated recurrent unit (GRU) [37], edge conditioned convolution (ECC) [38], and DenseNet [39] to construct the deep graph convolutional neural networks for the classification of super points. Super points without labels are to be ignored in loss computation.

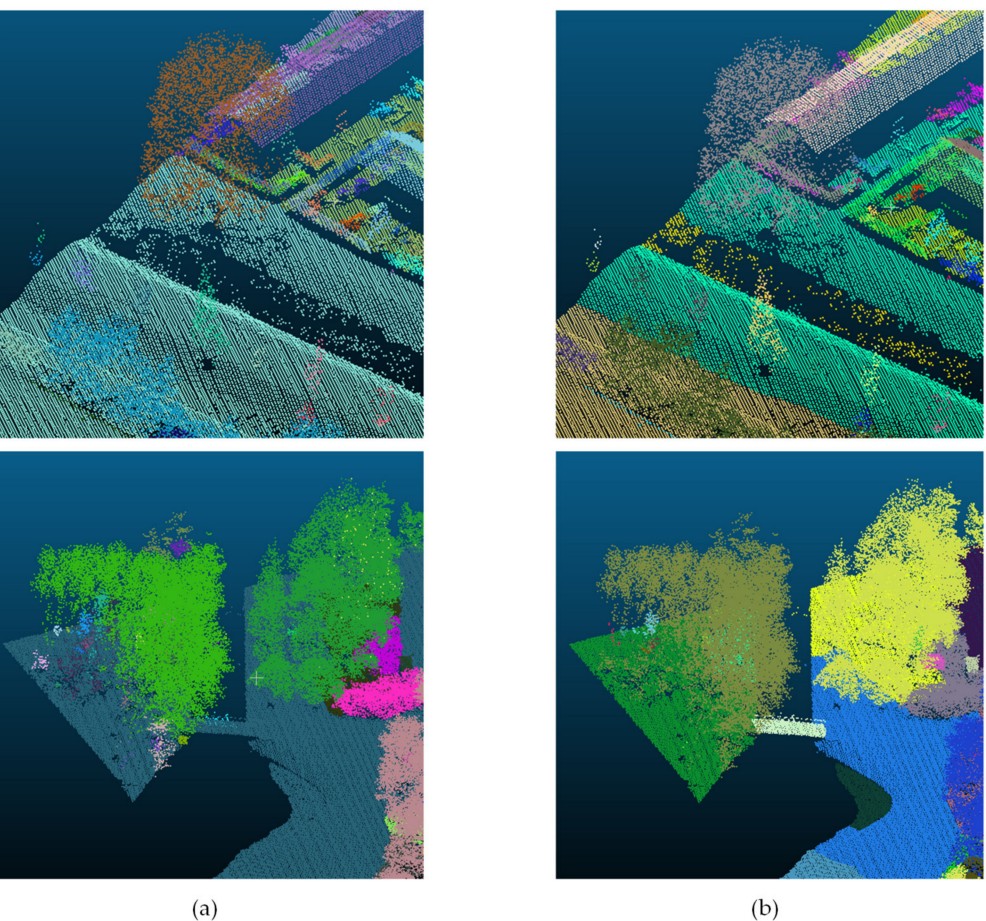

(a)                                               (b)

**Figure 5.** (**a**) Segmentation without intensity; (**b**) Segmentation with intensity.

## 4. Experimental Results

### 4.1. Dataset

The experimental area is in the Rotterdam city, Netherland. ALS point clouds are obtained from the Actueel Hoogtebestand Nederland v3 (AHN3 [40]). AHN3 is the digital height map for the whole of the Netherlands. It contains detailed and precise height data with an average of eight height measurements per square meter. Several products have been made of the measured heights, which can be roughly divided into two categories, i.e., grids and 3D point clouds. For the point clouds, AHN3 has point density >10 points/m$^2$. In addition, extra attributes such as intensity, the return number, and classification labels are included for each point. As for classification labels, each point is assigned to one of the following classes by manually labeling: terrain, water, building, artwork (mostly bridge points in our experiment), and others (mostly tree points in our experiment). The size of the experiment area in Rotterdam city is $5 \times 6.25$ km, as shown in Figure 6a. This region is divided into four $2.5 \times 3.125$ km parts, three of them are used for training (contains 417 million points) and the remaining one is used for testing (contains 140 million points).

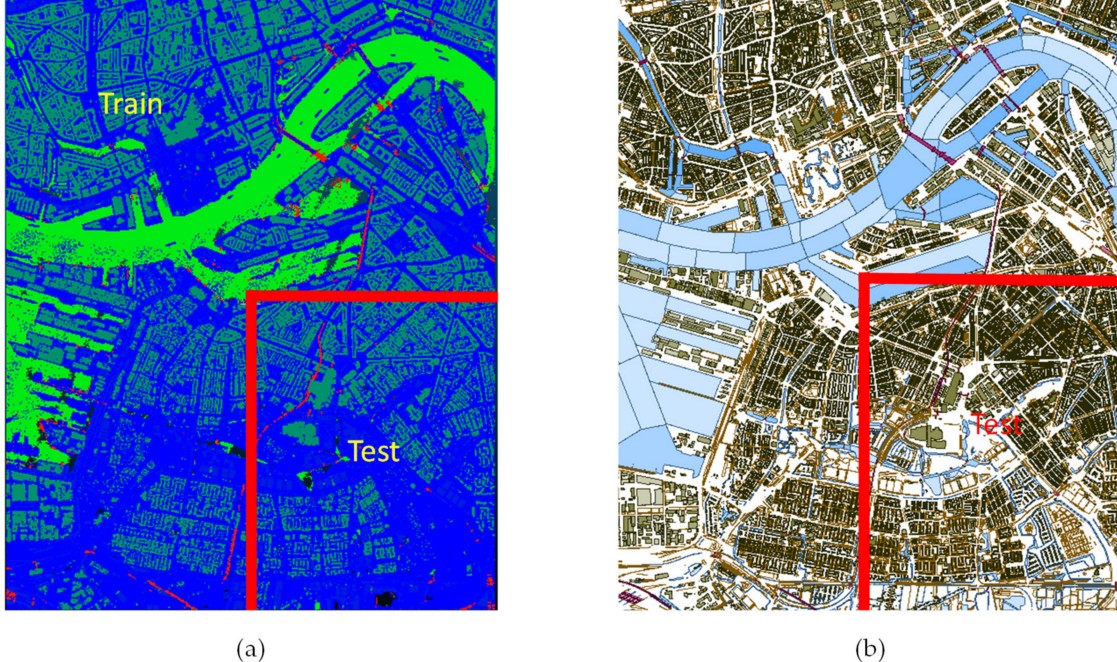

|            | (a)            |            | (b)        |

**Figure 6.** Experiment data in Rotterdam city. (**a**) AHN3 point clouds data; (**b**) Four classes derived from the BGT topographic map (polygons from the testing area are not used in the prediction work).

The topographic maps are obtained from the Basisregistratie Grootschalige Topografie (BGT [41]) data. BGT is a digital map of the Netherlands on which buildings, roads, watercourses, sites, and railway lines have been recorded. The map is accurate to 20 centimetres and contains many details. As shown in Figure 6b, four types of polygon maps (bridge, water, vegetation, and building polygons) derived from the BGT topographic map are used in our experiment.

The point-in-polygon operation is applied to the AHN3 and BGT data. Table 1 shows the covariance matrix between the ground truth results (from AHN3 labels) and the point-in-polygon operation results for the training area. In Table 1, values in the "Num" column show the number of ground truth points in the corresponding category. Other values in each row indicate the proportion of point-in-polygon labels in the corresponding category in (%). In the training area, 47.2% of tree points remain unlabeled and 28.9% of tree points remain unlabeled in the testing area.

**Table 1.** Confusion matrix between the ground truth results and the point-in-polygon operation results for the training area. GT indicates the ground truth and PiP indicates the point-in-polygon operation. None, indicates unlabeled points and Num, indicates the number of the ground truth points.

| GT＼PiP | Tree | Terrain | Building | Water | Bridge | None | Num |
|---|---|---|---|---|---|---|---|
| Tree | 48.0 | 0.3 | 4.1 | 0.1 | 0.3 | 47.2 | 103286913 |
| Terrain | 0.3 | 97.5 | 0.9 | 0.1 | 0.5 | 0.7 | 134736757 |
| Building | 0.6 | 0.3 | 92.9 | 0 | 0.1 | 6.1 | 150055441 |
| Water | 0.1 | 1.4 | 0 | 98.4 | 0 | 0.1 | 23445286 |
| Bridge | 4.9 | 16.8 | 5.7 | 0.6 | 60.5 | 11.5 | 5613662 |

### 4.2. Influence of the Unsupervised Segmentation

The initial point-in-polygon training samples still contain some unlabeled and mislabeled points. The unsupervised segmentation method is used to modify the initial training samples. To evaluate this improvement, a state-of-the-art point-based deep neural network Pointnet++ is applied to the initial training samples and comparing with our segmentation based strategy.

The AHN3 point clouds are divided into blocks to train and test the network. Our method is trained from scratch. The point clouds block should contain as much feature information as possible. Considering the calculation efficiency of the segmentation and SPG [32], the size of each block is $250 \times 250$ m. We adjust the segmentation result by adjusting the restrict parameter $\mu$ for further graphic neural network classification. The network is trained for 500 epochs. The learning rate starts from 0.02 with a decay rate of 0.7. The training steps of the network are (350, 400, 450). To evaluate the classification result, the overall accuracy and *F1* score are used to assess the performance of our resulting values. In the benchmark, the *tp*, *fp*, and fn are true positive, false positive, and false negative, respectively, and can be calculated using pixel-based confusion matrices per tile or an accumulated confusion matrix. In each class, we compute the precision and recall as follows:

$$precision = \frac{tp}{tp + fp}$$

$$recall = \frac{tp}{tp + fn} \tag{4}$$

Then, the *F1* score is calculated as

$$F1 = 2 \times \frac{precision \times recall}{precision + recall} \tag{5}$$

The overall accuracy is the normalization of the trace from the confusion matrix.

Figure 7 shows the performance of our framework on training area. Figure 7a shows the part of our initial point-in-polygon labels, some of the tree points are mislabeled into building category since the occlusion problem, as mentioned in Section 3.1. Figure 7b shows the corresponding segmentation result. Figure 7c shows the classification result by SPG using the training samples after the unsupervised segmentation refinement. In the testing area, training samples generated from the training area are used to train the SPG, the topographic maps are not used in this classification step. Table 2 shows the SPG results for the testing area using different segmentation restrict parameter $\mu$. Each column shows the *F1* scores for each category, the overall accuracy, and average *F1* score under corresponding restrict parameter $\mu$ in (%). At $\mu = 0.2$, we obtained the best average *F1* score (65.6%).

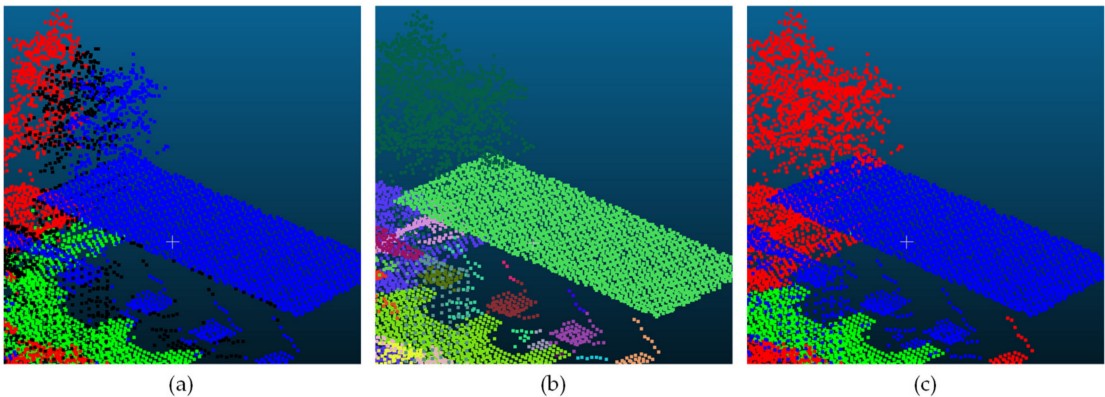

(a)                          (b)                          (c)

**Figure 7.** Performance of our framework in the training area. (**a**) Initial point-in-polygon labels; (**b**) Unsupervised segmentation result; (**c**) Classification result by SGP. Building, navy blue; tree, red; terrain, green; unlabeled, black.

**Table 2.** SPG results for the testing area. The *F1* score is used to evaluate the performance. Bold numbers show the highest values within different parameters.

| Class | $\mu_{0.1}$ | $\mu_{0.2}$ | $\mu_{0.3}$ | $\mu_{0.4}$ |
|---|---|---|---|---|
| Tree | **87.8** | 85.7 | 86.1 | 86.8 |
| Terrain | **92.3** | 92.0 | 91.3 | 90.5 |
| Building | **91.1** | 89.2 | 87.7 | 88.5 |
| Water | **5.8** | 2.0 | 1.1 | 1.2 |
| Bridge | 43.0 | **59.2** | 28.2 | 35.7 |
| Overall Accuracy | **90.0** | 88.6 | 88.1 | 88.3 |
| Average *F1* | 64.0 | **65.6** | 58.9 | 60.5 |

To train the Pointnet++ [24], the point clouds are divided into regular blocks. Different from the SPG, Pointnet++ is a point-based classification method and has a limit receptive field over the 3D scene. Considering the calculation efficiency of the Pointnet++, the size of each block is $50 \times 50$ m. In each block, 20,000 points are treated as input and sampled using the iterative farthest point sampling strategy. If the block contains less than 20,000 points, then, we duplicate these points because the Pointnet++ can only take a fixed number of points in each point cloud block. To suit the AHN3 dataset, four levels are contained in Pointnet++. The number of points at each level are (4096, 1024, 256, 64). Each level contains two scales, i.e., 16 neighbors and 32 neighbors. The search radius of 16 neighbors at each level are (2, 4, 8, 16) meters. The search radius of 32 neighbors at each level are (4, 8, 16, 32) meters. The Pointnet++ network is trained for 70 epochs. During the training, the learning rate starts from 0.005 with a decay rate of 0.7 at every 5 epochs. The learning rate stops decreasing when it is smaller than 0.0001 and its value remains at 0.0001.

The classification results of Pointnet++ are compared with the best segment-based strategy results ($\mu = 0.2$, highest average *F1* scores) as shown in Figure 8. Our framework has a better performance on average *F1* score (65.6%) than the point-based method Pointnet++ (59.4%).

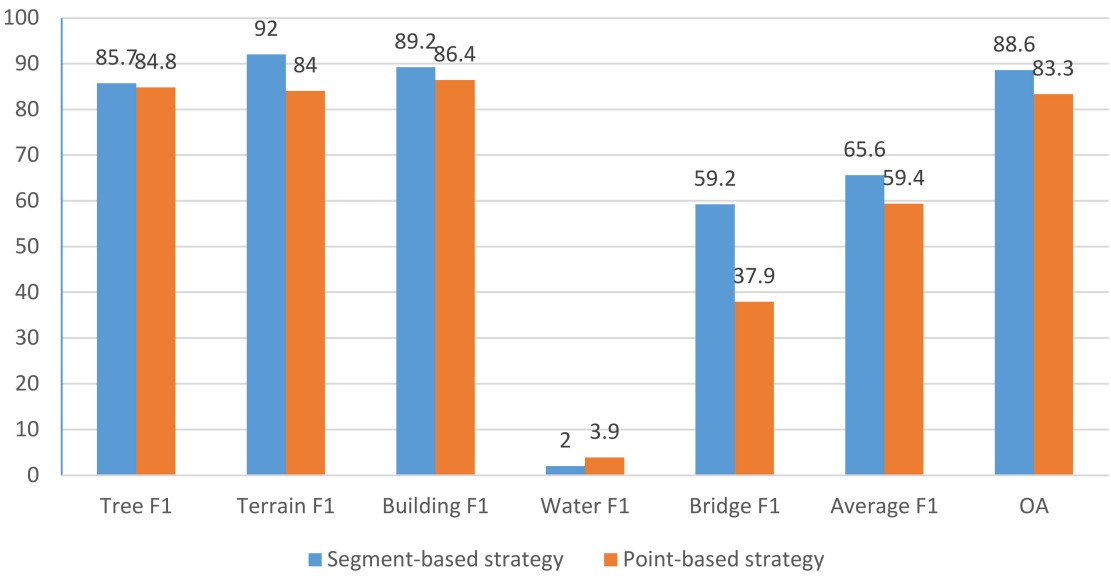

**Figure 8.** Comparison between segment-based strategy and point-based strategy in the testing area using initial point-in-polygon training samples.

### 4.3. Influence of the Intensity

To solve the problem shown in Figure 5a, and further improve the performance of our initial training samples, we add intensity values into the segmentation procedure. The influence of this strategy on the classification result is shown in Figure 9. As shown in Figure 9b, water and bridge can be classified after adding the intensity value in the segmentation. Table 3 shows the classification results for the testing area using a different segmentation restrict parameter μ, after adding intensity value into the segmentation procedure. At μ = 0.3, we get the best average *F1* score (74.8%).

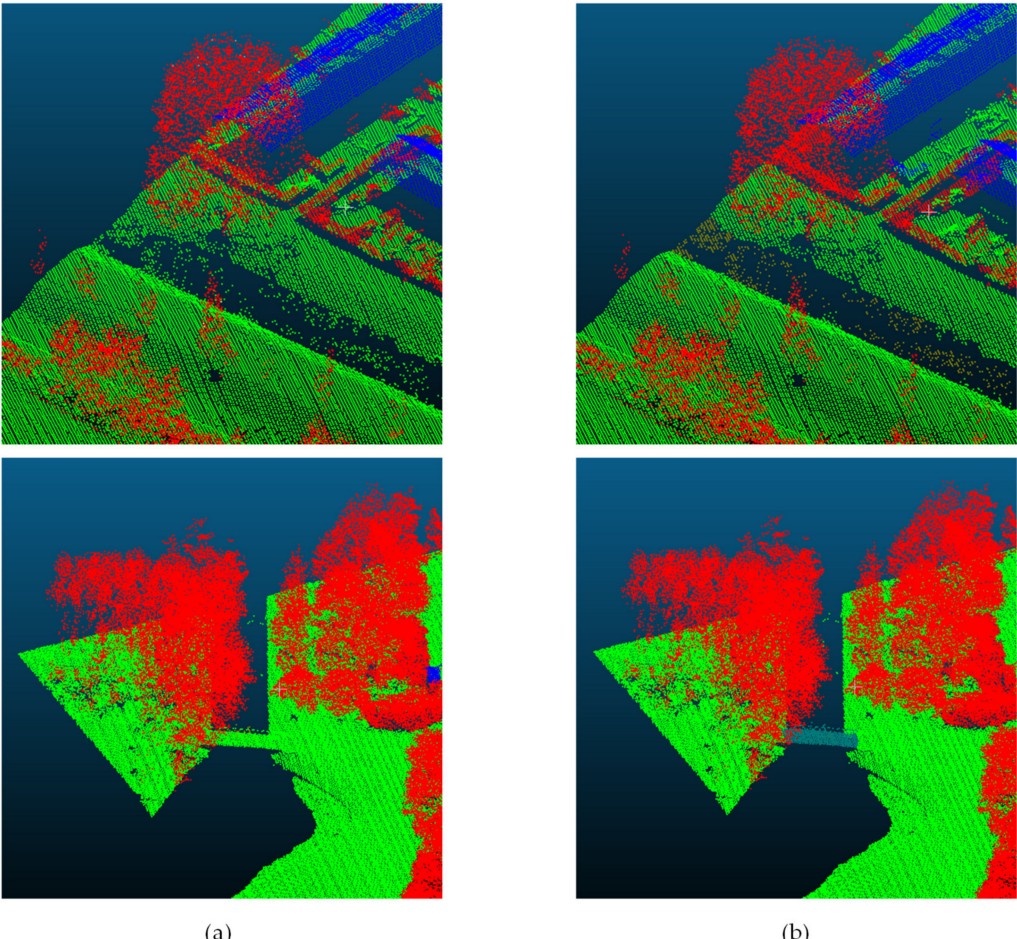

(a)　　　　　　　　　　　　　　(b)

**Figure 9.** (**a**) Classification results without intensity; (**b**) Classification results with intensity. Building, navy blue; tree, red; water, brown; terrain, green; and bridge, cyan.

**Table 3.** Results for the testing area after adding intensity values. The *F1* score is used to evaluate the performance. Bold numbers show the highest values within different parameters.

| Class | $\mu_{0.1}$ | $\mu_{0.2}$ | $\mu_{0.3}$ | $\mu_{0.4}$ |
|---|---|---|---|---|
| Tree | **89.0** | 85.8 | 88.6 | 87.3 |
| Terrain | **92.8** | 90.3 | 91.6 | 91.4 |
| Building | **93.2** | 92.2 | 91.4 | 89.4 |
| Water | 20.6 | 25.4 | 39.7 | **41.6** |
| Bridge | 39.0 | 48.3 | **62.7** | 44.8 |
| Overall Accuracy | **91.1** | 88.9 | 90.2 | 89.0 |
| Average *F1* | 66.9 | 68.4 | **74.8** | 70.9 |

*4.4. Comparison of the Result on Different Training Samples*

Training samples generated from the proposed framework are compared with the ground truth training samples. The comparison is based on the impact of the different training samples on the corresponding classification results, as shown in Figure 10. After a proper unsupervised segmentation, we get the best classification result using the proposed framework ($\mu = 0.3$, highest average *F1* scores). The average *F1* score is 74.8%, and the overall accuracy is 90.2%. For the result using ground truth training samples, the parameters for segmentation remain unchanged and we use the same deep networks. The average *F1* score is 75.1% and the overall accuracy is 93.2%.

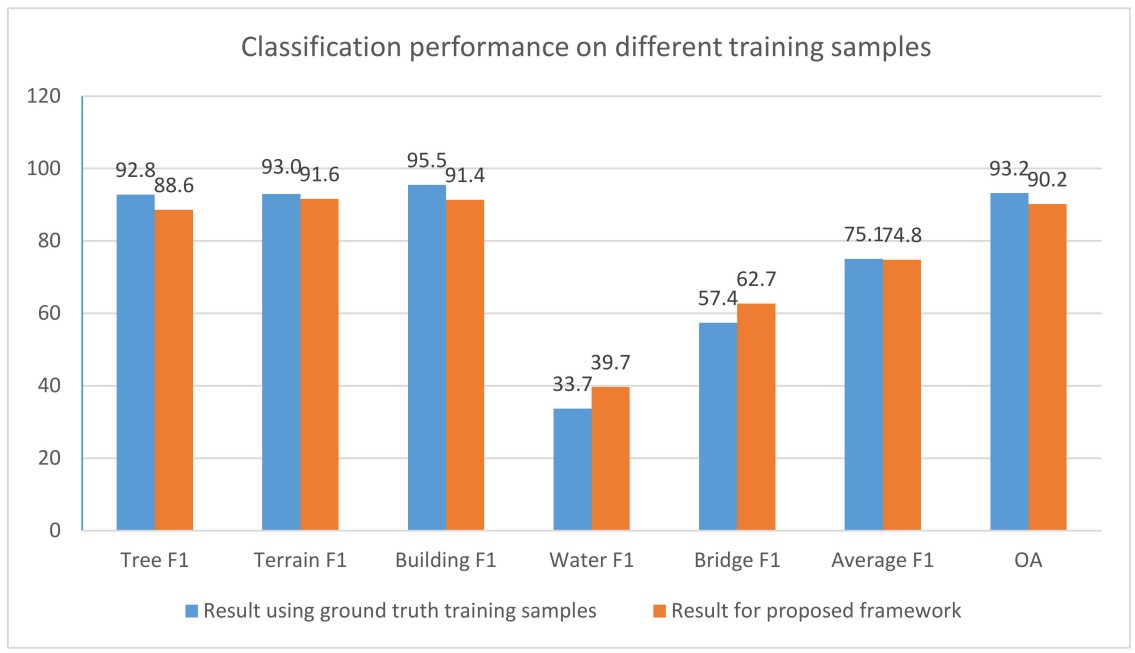

**Figure 10.** Classification results for the testing area using different training samples.

## 5. Discussion

A comparison with the ground-truth label, after the point-in-polygon operation, shows that most of the points have correct labels, as shown in Table 1. In the training samples, the recall values for terrain (97.5%), building (92.9%), and water (98.4%) have a good performance. Some parts of tree (47.2%), building (6.1%), and bridge (11.5%) points are unlabeled, as mentioned in Section 3.1. In addition, 16.8% of bridge points are mislabeled into terrain due to the inaccuracy of the DEM filter, 4.1% of tree points are mislabeled into the building due to the occlusion problem, as shown in Figure 7a. The unsupervised segmentation method is used to modify our initial training samples. After the segmentation, points belonging to one segment have the same label. As shown in Figure 7b, points over the building roof belong to the same segment as the rest of the tree points. Thus, the mislabeled problem is modified. The super point graph can be trained by these refined segments and do the final classification work. As shown in Figure 7c, most of the tree points are in the correct result, although they are mislabeled in the initial training samples.

The classification accuracy is highly correlated with the segmentation performance. By adjusting the parameter $\mu$, the best classification result is achieved ($\mu = 0.2$, highest average *F1* scores). As shown in Table 2, in the testing area, tree (85.7%), terrain (92.0%), and building (89.2%) points have good performances on *F1* scores. Adjusting the restrict parameter $\mu$ can separate the bridge from the terrain and building in the segmentation step. Thus, the *F1* score of the bridge improves from 28.2% to 59.2%. The *F1* score of water (2.0%) is low, since water and terrain points are adjacent in space and have similar geometric features, as mentioned in Section 3.2. Comparing with the point-based method

Pointnet++, using the same initial training samples, our framework has better results on both average *F1* score (+6.2%) and overall accuracy (+5.3%). Figure 11a,b shows part of the prediction result by our framework and Pointnet++. Some of the building facade points are misclassified into tree in Pointnet++ results. In our results, these points are correctly classified. These prediction errors could due to the mislabeled points in the initial point-in-polygon training samples. After the unsupervised segmentation, as shown in Figure 11c, the whole building facade is in one segment. The same situation occurs in our training samples. Since we assign the most frequent label to each segment, errors in the initial training samples are modified. Thus, our segment-based strategy has better performance than the point-based method.

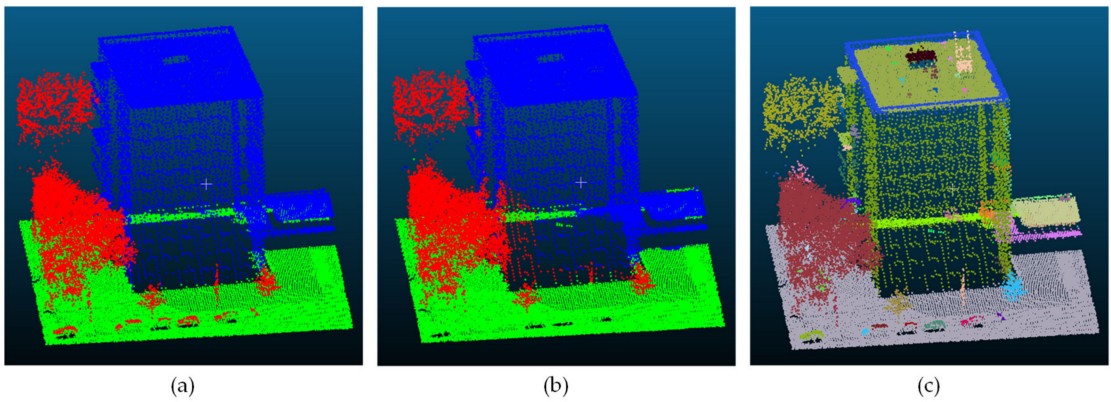

(a)                                            (b)                                            (c)

**Figure 11.** (**a**) Prediction result by our framework in the testing area; (**b**) Prediction result by Pointnet++ in the testing area; (**c**) The segmentation result. Building, navy blue; tree, red; and terrain, green.

Our framework has low performance on water, since it is hard to distinguish water points from terrain points only by geometric features in our experiment area. As mentioned in Section 3.2, the intensity value is introduced to solve this problem. Intensity value makes the segments geometrically simple and semantically homogeneous. Without the intensity value, points in small brook and bridge are clustered into the same segment with terrain points. Since each segment is assigned to only one label in our modified training samples, water and bridge points can be misclassified into terrain, as shown in Figure 9a. After adding intensity value into the unsupervised segmentation, water and bridge points can be separated from nearby terrain points. As shown in Figure 9b, points belonging to the small river and bridge can be and classified correctly. By adjusting the restrict parameter, we get the best result for testing (in Table 3, $\mu = 0.3$, highest average *F1* scores). The best average *F1* score improves from 65.6% to 74.8%. In Tables 2 and 3, the best *F1* score for water improves from 5.8% to 41.6%. The best *F1* score for bridge improves from 59.2% to 62.7%.

To evaluate the performance of the proposed framework, our best classification result is compared with the result using ground truth training samples. During the comparison, the parameters of the unsupervised segmentation and the super point graph remain unchanged. As shown in Figure 10, in the testing area, the average *F1* score for our framework (74.8%) has a competitive performance as compared with the result using ground truth training samples (75.1%). The result using ground truth training samples outperforms our framework result on tree class, since almost half of the tree points (47.2%) remained unlabeled in our initial training samples, as shown in Table 1. The result using ground truth training samples has more data to learn in tree class. Thus, our framework has worse performance on overall accuracy (−3.0%). However, on water (+6.0%) and bridge (+5.3%), our result has a better performance. Although using our framework to generate training samples can cause some points to be unlabeled, it can also reduce the noise effects. Thus, our automatically generated training sample framework has a similar performance on the average *F1* score as that of the result using ground truth training samples. The proposed framework has the ability to generate training samples automatically and do the classification work.

Additionally, our framework has the potential for further applications such as change detection. Objects could have been changed between the acquisition of the topographic map and the point clouds. In our experiment, the ALS data is acquired after the 2D topographic maps. Thus, some parts of the building and the tree points remain unlabeled using our point-in-polygon operation, as shown in Figure 12a. After the classification by our framework in the training area, the unlabeled building and tree points are correctly classified, as shown in Figure 12b. A comparison of the initial point-in-polygon label with the prediction result shows that the proposed framework can do the change detection work.

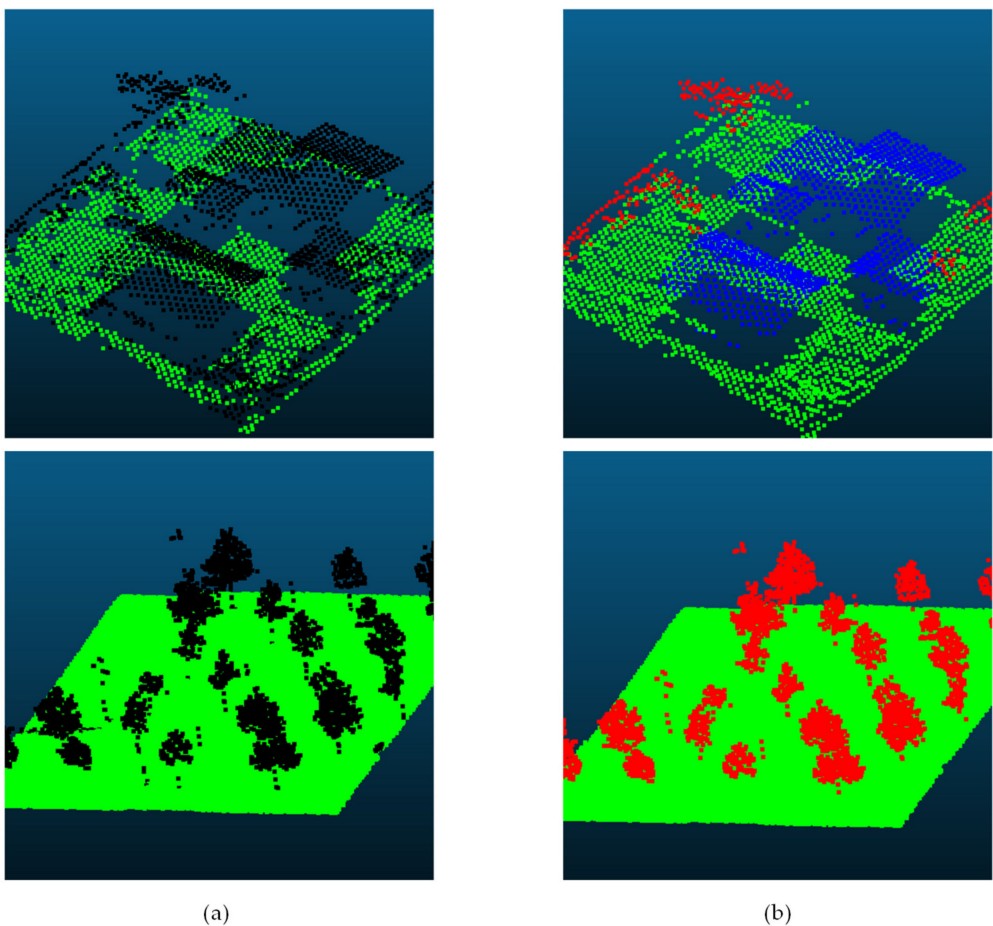

(a)　　　　　　　　　　　　　　　　　　　　　　　(b)

**Figure 12.** (**a**) The point-in-polygon labels on training area; (**b**) Prediction result by our framework in the training area. Building, navy blue; tree, red; and terrain, green.

Although adding intensity value solves the problem at the segmentation step, the *F1* score on water (41.6%) is still underperforming. As shown in Figure 13, the water scales vary a lot in the training area and the testing area. In the training area, water points are always in the big river and lake. At the segmentation level, these points are in the large and wide segments. However, in the testing area, brook contains most of the water points. Water points in the testing area can be in the small and thin segments. This can lead to misclassification in the testing area. To make full use of the proposed framework, a new method that can automatically select the most suitable training samples should be applied in our future work.

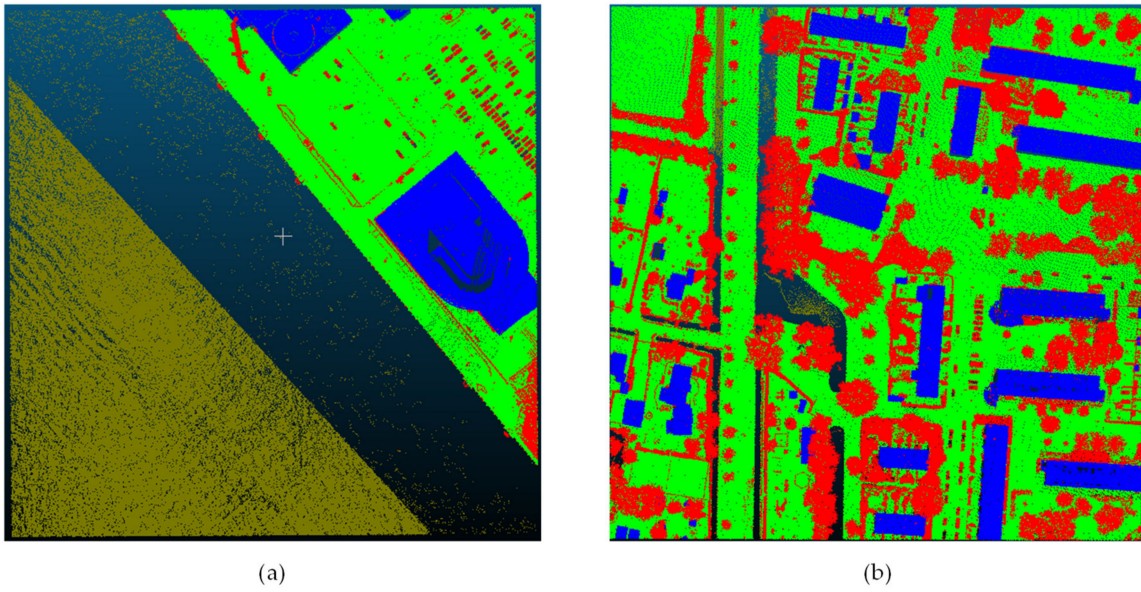

(a)                                                                                                (b)

**Figure 13.** (**a**) Water in the training area; (**b**) Water in the testing area. Building, navy blue; tree, red; water, brown; and terrain, green.

## 6. Conclusions

In this paper, we propose a framework that uses 2D topographic maps and unsupervised segmentation to automatically generate training samples. Using the domain knowledge through the topographic maps and the segmentation modification, training samples are generated from initial ALS data. The super point graph is trained by our automatically generated training samples and can do the prediction work. The framework is "unsupervised" in a computer vision sense that the classical ground truth training data are not needed. Our method still has the potential for improved performance. The information on the 2D topographic map is not fully utilized. Only five categories in ALS point clouds are labeled in our framework. In future work, more categories and more sources of data should be contained in our framework for a wider range of applications.

**Author Contributions:** Methodology, Z.Y., W.J., and S.O.E.; Validation, Z.Y. and Y.L.; Writing—original draft, Z.Y.; Writing—review and editing, S.O.E. All authors have read and agreed to the published version of the manuscript.

**Funding:** This research was funded by China Scholarship Council, grant number 201906270225. This work was supported in part by the National Key R&D Program of China, grant number 2018YFB0504800 (2018YFB0504801).

**Acknowledgments:** The authors gratefully acknowledge financial support from the China Scholarship Council.

**Conflicts of Interest:** The authors declare no conflict of interest.

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
