# Peer review of "Using Training Samples Retrieved from a Topographic Map and Unsupervised Segmentation for the Classification of Airborne Laser Scanning Data"

_remotesensing, doi:10.3390/rs12050877_

Round 1

Reviewer 1 Report

The authors present an unsupervised (in the sense there is no need for labelled point cloud training data) ALS point cloud classification framework with the goal to create consistent training datasets. The methodology leverage 2D topographic maps to automatically classify the underlying ALS point cloud. To study the fit of this classified dataset, it is used with deep learning architecture ad compred with ground truth training results.

The idea is very interesting and is surely a common community effort to find new and improve ways toward the creation of high-quality training datasets. However, the article presents many flaws, and suffer from a lack of focus toward the novelty, especially looking at previous works from same author(s) dealing with point cloud classification using topographic maps.

Main concerns :

The article suffers from a very dense style which is quite confusing for the reader. On top, there is an extensive amount of rewriting need as the English is either approximative or wrong, thus highly recommending an in-depth review of the article The methodology is more a collage of existing approaches rather than a real novelty. While it is an interesting application of the point cloud classification from 2D data from previous works, the justification of a paper in a journal aimed at scientific novelty is arguable. It seems a conference would be better suited. The methodology and experimental bench is quite confusing. It is quite unclear why the authors just focused on looking at the training scores (which is usually skipped in ML and Deep learning), looking only the results on the testing sets. This may be due to point (1) and a misunderstanding due to the writing style Authors confuse segmentation, classification, unsupervised / supervised, semantic …. And it should be clarified as well. How well does the approach compares against other “unsupervised” approaches ?

To go more in details

Abstract

The abstract in its current form is quite efficient. It is well structured and clear. However, some English mistakes should be addressed.

14           Semantically labelling è The labelling or semantic injection

Introduction

The introduction is quite lengthy and very dense. For a better quality, I suggest the author to dropping faster to the research question and challenge they aim to solve. As such, an estimate of 1 page should be enough to drive with a higher focus the clarity and pertinency of the introduction to the reader. Also, as constructed, the introduction is more a related work section, and should be restructured. The description of reference is not very adequate and feels like a collage of condensed “abstracts” from different papers, without a real added-value to the reader.

There should be a space before references.

Several references are missing, especially addressing unsupervised segmentation toward classification. E.g [1–6]

Engelmann, F.; Kontogianni, T.; Schult, J.; Leibe, B. Know What Your Neighbors Do: 3D Semantic Segmentation of Point Clouds. In Proceedings of the European Conference on Computer Vision (ECCV); Munich, Germany, 2018. Engelmann, F.; Kontogianni, T.; Hermans, A.; Leibe, B. Exploring Spatial Context for 3D Semantic Segmentation of Point Clouds. In Proceedings of the International Conference on Computer Vision (ICCV); IEEE: Istanbul, Turkey, 2018; pp. 716–724. Poux, F.; Billen, R. Voxel-Based 3D Point Cloud Semantic Segmentation: Unsupervised Geometric and Relationship Featuring vs Deep Learning Methods. ISPRS International Journal of Geo-Information 2019, 8, 213. Hu, S.M.; Cai, J.X.; Lai, Y.K. Semantic Labeling and Instance Segmentation of 3D Point Clouds using Patch Context Analysis and Multiscale Processing. IEEE Transactions on Visualization and Computer Graphics 2018, 14, 1–1. Papon, J.; Abramov, A.; Schoeler, M.; Worgotter, F. Voxel cloud connectivity segmentation - Supervoxels for point clouds. In Proceedings of the Computer Vision and Pattern Recognition (CVPR); IEEE, 2013; pp. 2027–2034. Tosteberg, P. Semantic Segmentation of Point Clouds using Deep Learning, 2017. Qi, C.R.; Yi, L.; Su, H.; Guibas, L.J. PointNet++: Deep Hierarchical Feature Learning on Point Sets in a Metric Space. In Proceedings of the Conference on Neural Information Processing Systems (NIPS); Long Beach, United States, 2017. Serna, A.; Marcotegui, B. Detection, segmentation and classification of 3D urban objects using mathematical morphology and supervised learning. ISPRS Journal of Photogrammetry and Remote Sensing 2014, 93.

Materials and Methods

This part is really unclear, and the main figure doesn’t really help or bring decisive value to the comprehension. While the proposed approach is quite simple in essence, at this point it seems “serialized” and what happens if the first step in the pipeline fails ? Can the system recover ?

Also, did you address the possibility of georefencing errors ? What are the thresholds and different injected knowledge that are necessary in order to obtain a consistent initial classification ?

In general, I believe the methodology should be pushed deeper and address the underlying questions with more depth before moving to the “straight-forward” benchmark by using pre-established DL models.

The general feeling at this step is that the methodology is somewhat not the core of the article (thus in contradiction with the aim of any scientific journal which wants to bring real scientific novelty)

Results

The result section is a little “hashed” which is not really fluid for the reader. A first improvement would be to fluidify the section. Secondly, the point by point analysis of the result is arguable, but I suggest the author better defining what is worth being in the main text and what is to move to annexes (example, the different confusion matrix are not really important at this step).

Why did the author didn’t compare with other unsupervised results ?

Also, the score part is not really clear as how did the authors conducted their experiments. Which parameters are fine-tuned ? why ? what are the performances ? …

Discussion

Again, the writing style may not due justice to the aim of the authors.

While the initial idea to use results of “unsupervised” workflow to create datasets for supervised workflow is interesting, the discussion doesn’t quite emphasize the results in this direction. In a more transversal thinking, listing future works and a research agenda is interesting (should be taken out of conclusion for this part) but should be clearly supported by the current experiments. Also, the authors emphasize too much on the DL architecture and specificities rather than the serialized workflow presented in the section 2.

Conclusion

The conclusion is too close to the abstract and presents many English mistakes, it should be rewritten.

The paper in its current form presents too many flaws and an extensive amount of rework to be consider for publication. More importantly, the scientific novelty and soundness of the experiments is not of the highest standard and are subject for large improvement for a publication in a journal.

Nevertheless, I would like to highlight the effort and research direction of the team toward the extraction of knowledge from topographic maps for point cloud classification.

Author Response

Dear Professor:

Thank you for reviewing our manuscript entitled, “The classification of Airborne Laser Scanning data using training samples retrieved from a topographic map”, and giving review comments immediately.

We have carefully read your review comments and revised the manuscript. And the responses corresponding to your comments are listed as following.

Thanks again and appreciate your time to process this manuscript.

Sincerely yours,

Reviewer 2 Report

General comments:

This paper presented a methodology to improve training samples automatically obtained considering the use of a 2D topographic map and intensity values. The paper sections are well organized. The background of the study, its aims, existing literature and method description were addressed by the authors. I believe that the proposed paper is interesting and it can contribute to ALS data users. Therefore, I recommend this paper for publication. However, the novelty and the contribution of the manuscript are rather presented. I suggest to improve the introduction to highlight the contribution, advantages and disadvantages of the proposed method compared to the related works presented.  Some sentences in the manuscript can be improved for the seeking of clariness. Spoke English need to be avoid and a grammar revision is welcome. In terms of results and discussion, the discussion section should be carefully improved. The reason of some mislabeled and unlabeled points could be better addressed and suggestions to improve the method limitation are missing.

Title, Abstract and Introduction

Title: I suggest to improve the title to valorize the work. Is the use of a topographic map to obtain training samples the main contribution of the paper? Which is the main novelty presented in this work compared to previous works?

The abstract is well organized. However, I suggest to improve the method description from Line 18 to line 24. The novelty/contribution of the method need to be highlight. The comparative analyses with other methods can be mentioned and the results presented need to be related with the ground truth.

p.1 Line 35-In the first paragraph: “play an important role” is used two times (in line 35 and line 38). The word “caused” maybe is not the best choose to open the introduction. So, I suggest to rephrase the first sentence. A small revision in the first paragraph is required.

Citations: I noticed that just the first author’s names were used in the most part of the citations along the manuscript. When the cited paper has two or more authors “et al.” need to be used. See instruction for Authors. Example: And add the note to the text together with the citation: … recently reported by Díaz et al. [10]. https://mdpi-res.com/data/mdpi_references_guide_v5.pdf

p.1 line 41- Even though the acronym was mentioned in the abstract, it needs to be presented again the first time mentioned in the main text. For instance, ALS. Please, double check it along the manuscript.

In the introduction a very nice overview of the related works is presented, which can be considered a contribution in the paper. However, the related works description and connection between them could be better explored. I suggest some small improvements in terms of organization and connection. The related works could be presented more chronologic (year of publication). For instance, references 9 to 12 could be reorganized in the paragraph, among others. Maybe some small sentences to connect the related works mentioned would be welcome. The advantages and disadvantages of the methods could be mentioned and compared to include some connection between the related works cited, showing how the ALS classification had been improved along the last ten years. Furthermore, sentences such as “improves the final results” (line p.2 69) and “competitive classification results” (p.2 - line 80) needs to be follow by some numerical/ quantified values, otherwise, the related work description can be considered too vague.

p.3 line 129. The author mentioned: “in other to solve this problem”. Which problem? Just, noise and background sensibility? I suggest to be more specific to highlight the contributions/advantages of the method proposed, especially relating to the related works described in the introduction.

Methodology and Dataset

p.4 - line 144 to 145- This sentence is strange. Please improve the English. “Five categories, contained in our training samples that are terrain, water, vegetation, building and bridge

p.4 - line 146 – I’m not sure if Figure 1 is really necessary. In my opinion, just the test description is enough to understand this method step. I suggest to remove Figure 1 or improve the workflow with more interesting details.

p.4 line 150 - “are contained”? Or belongs to the ground/ non-ground classification. The use of “part” is too informal. Please improve the sentence. I suggest to check/improve some spoke English sentence along the manuscript.

p.4 Section 2.1 - The topographic map and the ground truth (Figure 2) could be better explained/ described in the methodology. The topographic map is better presented in Section 3.1. Maybe this is can be mentioned or a general description of what type of topographic map is required for the presented methodology can be interesting. How the precision of the topographic map impact the method proposed? Which is the required precision of the topographic map? How this ground truth were obtained? And which is the accuracy of the ground truth?

p.4 Section 2.1 - The main reason for mislabels points was the nature of the feature? The point-in-polygon results are not so accurate in the vegetation border too (Figure 2). Can you give more details about the outdated and inaccurate polygon map? (line 164). Maybe some directions how you improve this results considering the intensity value could be included here to connect with the next section.

line 177. the (,) is misplaced in the sentence. Suggestion: “Simple shape categories, such as terrain or water, are in large clusters. Categories such as building and tree are in smaller components”.

p.5 186 – Ref 21 – (Landrieu, L. and Simonovsky). HIS paper is used. Please check the citation carefully again in the entire manuscript.

p. 8 line 252 - In the Deep Learning Methods used in the performance analyses, which criterion was used to define how point clouds were divided ( 250*250 for SPG and 50*50 for Pointnet++). How this threshold can influence the comparative analysis?

RESULTS AND DISCUSSION

p.13 -The authors mentioned that “The 2D topographic map and the corresponding ALS data may not be acquired at the same date, objects such as building can be unlabeled or mislabeled due to that reason.” This affirmation seems confuse for me. The data set may not been acquired at the same date? Or was it not acquired in the same date? How much is the acquisition time difference? How much a fix target, such as a building could change during this period? Considering the precision of the point cloud used, are you sure that it can interfere the classification? If yes. How this acquisition time difference affected non-fix target, such as vegetation?

p.13 -Line 338-339. Please improve the sentences in this paragraph.

Figure 7 show that building façade and roofs were mislabeled due to tree occlusions. There are lot of methods in the literature that indicates how to define occluded roof and building edges in ALS point clouds, especially occlusion caused by vegetation. The authors could include a discussion and some references/directions how this limitation can be improved.

line 363 – “the situation improves”? Which situation?

I suggest to highlight/ quantify how much is the improvement (%) proposed with the method compared to time processing and implementation effort. The authors should present the applicability/reproducibility of the proposed method/framework. How much it worth for possible users? Which are the main applications?

An overview of related works was presented. However, in the discussion/conclusion a comparison with this related works are not performed. I suggest to relate the improvement mentioned by the authors with the previous work. Which are the main advantages in the used of the proposed method?

Author Response

(The authors gave the same response as above.)

Reviewer 3 Report

This manuscript presents an interesting work.
In general, the manuscript should be acceptable for publication but some problems must be repaired prior to publication.
They are:

1. The references list needs to be updated. There are many studies using neural network-related methods in classification of ALS point clouds in recent years.
1. Page 1 Line 36-39: It would be useful to show how previous studies perform the classification tasks of ALS point clouds.
2. Page 5:“I suggest to describe the labeling colors for five categories in Figure. 2 (and other related figures).
3. Page 8 Line 252:“Why the AHN3 point clouds are divided into 250*250 253 blocks to train?
4. Page 13 Line 335:“It seems that the restrict parameter u needs to be adjusted for improving performance,the authors should explicitly explain if the labelling method could obtain the better results compared to the manual labelling?
5. Please use the journal format in the reference list.

Author Response

(The authors gave the same response as above.)

Round 2

Reviewer 1 Report

Dear authors,

Thank you for addressing almost all of the previously made comments.

The paper is now largely improved, and after several careful reviews I have only very minor suggestions.

In general, the paper is much clearer, and I expect readers to quickly grasp why your research is important.

Also, I reiterate the nice approach to leverage Topographic data within training dataset constitution.

Generally, some english artefacts remains. Also, it seems the authors could drop faster to key idea, by fine-tuning the writing style and swapping long paragraphs by much simpler get-to-the-point deas (minor).

Some questions remain.

How did you control the Ground truth "manual" generation ? Did you assess it against someone else ?

How do you see your methodology for datasets from MLS, TLS and Terrestrial photogrammetry ?

The figure 1 is not optimal, as you use "database-like" icons for data points, and rectangles with overlapping lines. It seems it can be more impactful, and maybe better translate the DL architecture that you used (SPG) ?

Some could argue that it is falsly "unsupervised", as you still inject domain knowledge through topographic map and parameters supervision. Maybe clarifying that you are "unsupervised" in the CV sense that you do not need classical point cloud training data.

I suggest to improve the conclusion which is a bit too lenghty and doesn't do perfect justice to your work.

Well done.

Author Response

Dear Professor:

Thanks for your kindness comments!

We have carefully read your review comments and revised the manuscript. And the responses corresponding to your comments are listed as following.

Thanks again and appreciate your time to process this manuscript.

Sincerely yours,
